# Early biochemical analysis of COVID-19 patients helps severity prediction

**Andrés Roncancio-Clavijo**[1,2◉], **Miriam Gorostidi-Aicua**[1◉], **Ainhoa Alberro**[1,2,3], **Andrea Iribarren-Lopez**[1], **Ray Butler**[4], **Raúl Lopez**[4], **Jose Antonio Iribarren**[5], **Diego Clemente**[3,4,5,6], **Jose María Marimon**[7], **Javier Basterrechea**[8], **Bruno Martinez** [8], **Alvaro Prada**[1,2‡], **David Otaegui** [1,2,3‡] *

**1** Biodonostia Health Research Institute, Neurosciences Area, Multiple Sclerosis Group, San Sebastian, Spain, **2** Osakidetza Basque Health Service, UGC Laboratories Gipuzkoa, Immunology Section, San Sebastián, Spain, **3** Centro de Investigación Biomédica en Red en Enfermedades Neurodegenerativas-Instituto de Salud Carlos III (CIBER-CIBERNED-ISCIII), Madrid, Spain, **4** Butler Scientifics S.L., Barcelona, Spain, **5** Infectious Diseases Department, Osakidetza Basque Health Service, Donostialdea Integrated Health Organization, San Sebastián, Spain, **6** Neuroimmune-repair Group, Hospital Nacional de Parapléjicos-SESCAM, Toledo, Spain, **7** Microbiology Department, Biodonostia Health Research Institute, Infectious Diseases Area, Respiratory Infection and Antimicrobial Resistance Group, Osakidetza Basque Health Service, Donostialdea Integrated Health Organization, San Sebastián, Spain, **8** Osakidetza Basque Health Service, UGC Laboratories Gipuzkoa, San Sebastián, Spain

◉ These authors contributed equally to this work.
‡ AP and DO have managed equally this work.
* david.otaegui@biodonostia.org

**Data Availability Statement:** All relevant data are within the paper and its Supporting Information files.

## Abstract

COVID-19 pandemic has put the protocols and the capacity of our Hospitals to the test. The management of severe patients admitted to the Intensive Care Units has been a challenge for all health systems. To assist in this challenge, various models have been proposed to predict mortality and severity, however, there is no clear consensus for their use. In this work, we took advantage of data obtained from routine blood tests performed on all individuals on the first day of hospitalization. These data has been obtained by standardized cost-effective technique available in all the hospitals. We have analyzed the results of 1082 patients with COVID19 and using artificial intelligence we have generated a predictive model based on data from the first days of admission that predicts the risk of developing severe disease with an AUC = 0.78 and an F1-score = 0.69. Our results show the importance of immature granulocytes and their ratio with Lymphocytes in the disease and present an algorithm based on 5 parameters to identify a severe course. This work highlights the importance of studying routine analytical variables in the early stages of hospital admission and the benefits of applying AI to identify patients who may develop severe disease.

## Introduction

COVID-19 is an infectious disease produced by the coronavirus SARS-CoV-2 (Severe Acute Respiratory Syndrome Coronavirus 2). The first outbreak was reported in a seafood market in Wuhan, (Hubei province, China) and the World Health Organization (WHO) declared the

**Funding:** This Project has been funded by Instituto de Salud Carlos III (COV20/00314) that include part of FEDER founds. AA is funded by Basque government through a post-doc Grant. The funders had no role in study design, data collection and analysis, decision to publish or preparation of the manuscript."

**Competing interests:** The authors have declared that no competing interests exist.

situation a public health emergency of international concern on January 30<sup>th</sup>, 2020 and as a pandemic on March 11<sup>th</sup>.

COVID-19 disease has a wide clinical spectrum. A relevant proportion of infected individuals is asymptomatic, while others develop from mild symptoms, such as low-grade inflammation, fever and cough, to severe symptoms, that include respiratory failure, metabolic acidosis, coagulopathy, septic shock and multiorgan failure [1]. Although it is a rapidly changing scenario, based on current data, most people infected by the SARS-CoV-2 virus present a mild disease. Besides, depending on the study, different proportions of severe and critical patients, as well as mortality rates have been reported. In the case of Spain, a work that included 4035 hospitalized COVID-19 patients from 127 centres, reported 18.5% intensive care unit (ICU) admission and one-month mortality of 28%, which raised to 42.4% among ICU patients [2]. Prediction of the clinical course of each individual becomes a priority to optimize both the personalized treatment of each patient and the use of the medical resources.

Several works have tried to identify risk and prognostic factors based on demographic, epidemiologic, genetic, and clinical variables [2–6], and several models have been proposed for mortality and severity prediction, but they present diverse outcomes and no consensus has been reached yet on their application [7].

With this in mind, in this work, we took advantage of the routine blood tests performed on all individuals on the first day of hospitalization and analyzed the results of 1,082 COVID-19 patients generating a predictive model, based on these first-day data, that allows us to efficiently predict the course of the disease. Our results also highlight the importance of the immature granulocytes and their ratio with lymphocytes as parameters to be studied.

## Methods

This study has been conducted at the Donostia University Hospital, which covers the region of Gipuzkoa (Basque Country, Spain), with a population of 720,000 inhabitants. Data of the first blood analysis after the COVID-19 diagnosis have been retrieved from the clinical records. The inclusion criteria were having a SARS-CoV-2 positive PCR between February and April 2020 and being admitted to the hospital. Demographic data and variables are presented in **Tables 1** and **2**. The data has been obtained fully anonymized. The data has been obtained anonymously and the follow-up has been done until 30 days after the discharge of the hospital. This retrospective chart review study involving human participants was in accordance with the ethical standards of the institutional and national research committee and with the 1964 Helsinki Declaration and its later amendments or comparable ethical standards. The Human Investigation Committee (IRB) of Hospital Universitario Donostia (PI2020076) approved this study.

The severity categorization was done *a posteriori* according to [8]. "Severe cases" were defined as the ones meeting any of the following criteria: (1) Respiratory distress (≧30 breaths/

**Table 1. Demographic and clinical data of the study cohort.** Percentages for the first column were calculated for the total number of patients (1082). The percentages shown for Females and Males were calculated with respect to the total number of patients in each of the conditions, and the p-value was calculated for these comparisons. IQR = Interquartile range. ICU = Intensive care Unit.

|  | All | Female | Male | p-value |
|---|---|---|---|---|
| Samples (%) | 1082 | 545 (50.4%) | 537 (49.6%) | 0.81 |
| Mean age (min-max) | 66.5 (10–99) | 66.5 (10–99) | 66.5 (17–95) | – |
| Severe cases (%) | 449 (41.5%) | 184 (41%) | 265 (59%) | < 0.001 *** |
| ICU admission (%) | 60 (5.6%) | 16 (26.7%) | 44 (73.3%) | < 0.001 *** |
| Deceased (%) | 140 (12.9%) | 54 (38.6%) | 86 (61.4%) | 0.004 ** |

**Table 2. The normal range of the study variables and the data from COVID-19 patients were compared.** Fold change was calculated when the median value of patients was above the maximum or below the minimum value of the normal range. IQR = Interquantile range.

| | Normal Range | COVID-19 (min-max) | Fold Change |
|---|---|---|---|
| Albumin (g/dl) | 3.7–5.1 | 3.9 (2.07–4.94) | - |
| Creatine (mg/dl) | 0.4–1 | 1.02 (0.27–13.5) | 1.02 |
| Urea (mg/dl) | 10–65 | 43.97 (8–201) | - |
| Triglycerides (mg/dl) | 35–135 | 129.04 (34–1265) | - |
| Creatine kinase (CK) (U/l) | 0–188 | 150.92 (20–2662) | - |
| ALT/SGPT (%) | 0–33 | 25.79 (2–395.5) | - |
| AST/SGOT (U/l) | 0–31 | 30.33 (7–503) | - |
| LDH (U/l) | 60–190 | 277 (110–2290) | 1.46 |
| Troponin T (TnT) (ng/l) | 0–14 | 20.32 (2–393) | 1.45 |
| **C-reactive protein (mg/l)** | **0–5** | **50.16 (0–410)** | **10.03** |
| **IL-6 (pg/ml)** | **0–7** | **25.64 (1.72–166)** | **3.66** |
| Prothrombin Index (%) | 70–140 | 93.58 (4–137) | - |
| Fibrinogen (mg/dl) | 200–400 | 530.15 (266–780) | 1.33 |
| **D-dimer (ng/ml)** | **0–500** | **1351.25 (180–21000)** | **2.7** |
| **Ferritin (ng/ml)** | **15–150** | **426.71 (6.7–4557.3)** | **2.84** |
| Procalcitonin (ng/ml) | 0–0.5 | 0.36 (0.01–18.51) | - |
| Lactate (mmol/l) | 0.3–2 | 1.37 (0.5–6) | - |
| Platelets ($10^3$/μl) | 140–400 | 215.69 (12–729) | - |
| Leukocytes ($10^3$/μl) | 3.8–10 | 7.07 (0.03–77.44) | - |
| Neutrophils ($10^3$/μl) | 1.6–7.5 | 4.5 (0–18.81) | - |
| Neutrophils (%) | 40–75 | 63.16 (0–95.6) | - |
| Eosinophils ($10^3$/μl) | 0–0.6 | 0.12 (0–1.85) | - |
| Eosinophils (%) | 0.5–7 | 1.75 (0–16.9) | - |
| Basophils ($10^3$/μl) | 0–0.2 | 0.03 (0–0.16) | - |
| Basophils (%) | 0–1.5 | 0.45 (0–2.1) | - |
| Immature granulocytes ($10^3$/μl) | 0–0.03 | 0.04 (0–0.97) | 1.35 |
| Immature granulocytes (%) | 0–0.5 | 0.55 (0–23.2) | 1.11 |
| Lymphocytes ($10^3$/μl) | 0.9–3.5 | 1.82 (0.01–65.51) | - |
| Lymphocytes (%) | 19–48 | 25.33 (0.9–88.9) | - |
| Monocytes ($10^3$/μl) | 0.2–0.9 | 0.58 (0.02–7.36) | - |
| Monocytes (%) | 3.05–12 | 8.96 (0.5–66.7) | - |
| Partial pressure of oxygen (mmHg) | 83–108 | 73.18 (14–271) | -1.13 |
| Oxygen saturation (%) | 95–99 | 92.94 (21.2–100.8) | -1.02 |

min); (2) Oxygen saturation≤93% at rest; (3) Arterial partial pressure of oxygen (PaO2)/fraction of inspired oxygen (FiO2)≦300mmHg. Those patients that did not meet any of the conditions were considered "Non-severe cases".

Continuous variables have been described by the median, minimum and maximum and interquartile range while categorical variables have been described by the number of cases and percentages. For the continuous variables, the comparison of the means has been done using T-test, when the variables were normally distributed, and Mann-Whitney-Wilcoxon (2 categories) / Kruskal-Wallis (>2 categories) tests for the non-normally distributed. For the categorical variables, Chi-squared and Cramér's V index test were applied. The fold change (FC) was calculated for the variables that had a mean value out of the normal range. For increased variables, the mean value was divided by the maximum value of the normal range, and in those

below the normal range, by dividing the minimum value of the normal range by the mean value. FC values >|2| and a p-value lower than 0.05 were considered relevant. All statistical analyses have been done using R (R.3.6.3) [9]. In the charts, p-values has been explained as follows: * → p<0.05, ** → p<0.01 and *** →p<0.001.

We used AutoDiscovery (Butler Scientifics) for a stratified feature selection and to generate our final classification model. Briefly, a data set of 42 features with the data of 1,082 patients was constructed and 37 of these variables were chosen for the analysis [33 variables identified plus age group, sex, severity and the Lymphocyte to Immature Granulocyte ratio (L/IG)]. This L/IG ratio has been included based in clinical observation and in the analysis of our data. The next step was an automated data exploration, where the association and correlation between the variables were analyzed (AutoDiscovery Strength Score) to avoid model overfitting and to identify the patient's features that were more associated with the predictive goal. In this specific case, the association of the variables with the severity variable was evaluated and one of the correlated pair of variables was removed, if any. To do that, AutoDiscovery selected the proper numerical method based on the data type and distribution of the variables assessed. Numerical tests applied in each case are Spearman's Rank Correlation, Variance Analysis (ANOVA one-way, U Mann-Whitney, and Kruskal-Wallis; normality was tested in these cases with D'Agostino/Pearson methods), and Cramer's V Contingency Index.

This massive data exploration process was carried out systematically for each patient group (stratum), considering only those correlations evaluated in a data subset with n≥5, two-tailed p < 0.05 and Spearman's rank correlation coefficient > 0.8 to be relevant.

Given the nature of this multiple testing method, a high significance threshold was calculated based on Benjamini-Hochberg method (False-Discovery Rate) to classify the rest associations.

The Feature Selection process, where, based on Recursive Feature Elimination (RFE) and the AutoDiscovery Strength Score, age decade and the Lymphocytes to Immature Granulocyte ratio were added as variables for the model.

The model was trained and evaluated using three models: Top-5, Top-10 and Top-18 (using the 5,10 and 18 most significant variables), plus Age group and L/IG. The training and evaluation processes included the utilization of multi-class classifiers, such as Decision Trees, Logistic Regression, Naïve Bayes, K-Neighbors, SVM and Random Forest. The Model Training step is described in **Fig 1**, using the 60% of the samples for training, a 20% for validating and the final 20% for the testing, where the most optimized accuracy is obtained. Finally, a Cross-Validation method (K-fold, K = 10) is applied to decrease overfitting impact. Sensitivity,

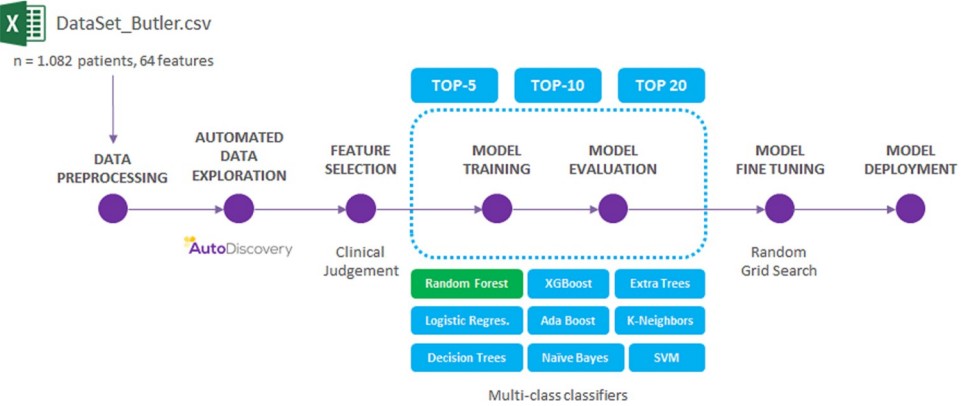

**Fig 1. Schematic representation of the classifier analysis.**

Specificity and accuracy has been calculated trough SciPy [10]. F1-score has been calculated from the Precision and the recall.

The results of all these steps are the Confusion Matrix, the accuracy and the ROC curve of each of the three Top models (S1 File).

## Results

The sex, age and clinical data of the 1082 participants are presented in **Table 1**. The incidence of severe cases, ICU admission and deceased patients was significantly higher among male individuals. A higher percentage of severe cases was reported with increasing age, while it is reduced in nonagenarians when compared to octogenarians (p-value = 0.04) (**Fig 2**).

The median value of most of the variables was in the normal range, only 4 of them have a fold change > |2|. Patients have more D-dimer, Ferritin, IL-6, and remarkably, 10.03 times more C-reactive proteins (**Table 2**).

To determine if any of these variables could be good predictors of disease severity, patients were grouped based on their severity outcome. Notably, as shown in **Table 3**, we found that 32 out of the 33 variables tested were significantly different between severe and non-severe cases. Eleven variables show lower values in severe group, 21 were increased. There was a strong statistical significance for most of the variables, with 26 variables with a p-value < 0.001. We also calculated the L/IG for all the patients. The obtained results demonstrate that there is a strongly significant reduction of the L/IG in the first blood analysis of patients that then develop a severe disease.

Since almost all variables appeared significantly different between severity groups, even if the mean value of most of them was in the normal range, it was decided to create a classification model to find the most relevant variables that would be able to predict the severity outcome of the patients at an early stage of the disease.

AutoDiscovery software was used for the creation of a predictive model. In the Automated Data Exploration step an association of almost all variables with severity variable was found. The ranking of the variables obtained in the Feature Selection step is shown in **Fig 3**, where the specific variables of each model are described.

Among all the multi-class classifiers, Random Forest showed the best performance. After the Cross-Validation process and the application of RFE, the results have shown an accuracy

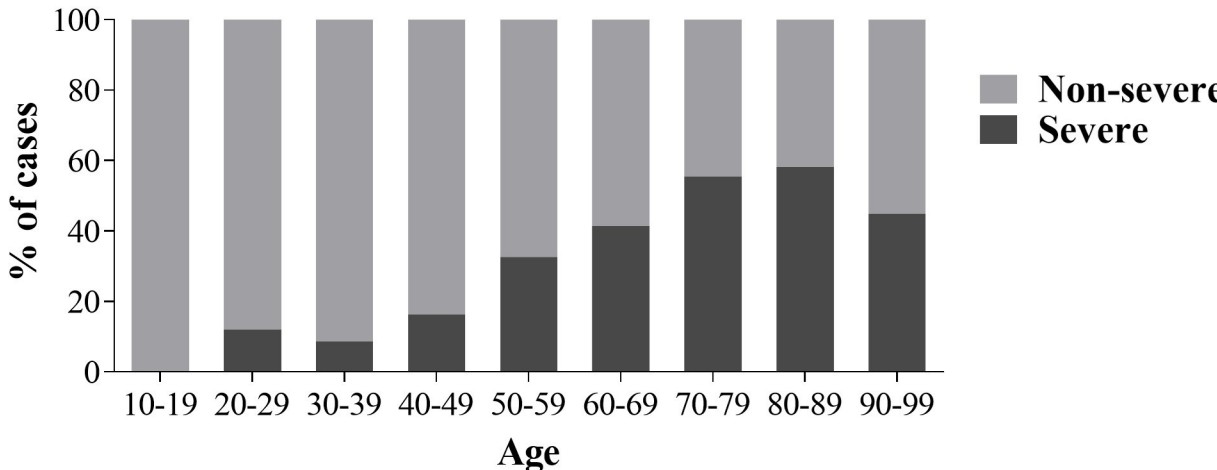

**Fig 2. Age distribution of COVID-19 severity.** For each of the age ranges the percentage of Non-severe and Severe cases are shown. Age is presented in years.

**Table 3. Comparison of the study variables between the COVID-19 patients that have a "Non-severe course" and the ones that develop a "Severe disease".** 32 of the 33 evaluated variables are significantly different. For an easier visual interpretation of the results, analytes with lower values in severe patients are shown in white and the analytes with higher values in gray. The "Sample n" column indicates the number of data available for each analyte. IQR = Interquantile range.

| | Normal Range | COVID-19 (min-max) | Non-Severe (min-max) | Severe (min-max) | p-value | Sample n |
|---|---|---|---|---|---|---|
| Albumin (g/dl) | 3.7–5.1 | 3.9 (2.07–4.94) | 4 (2.07–4.94) | 3.76 (2.15–4.82) | < 0.001 *** | 309 |
| Creatine (mg/dl) | 0.4–1 | 1.02 (0.27–13.5) | 0.92 (0.27–13.5) | 1.16 (0.29–10.44) | < 0.001 *** | 869 |
| Urea (mg/dl) | 10–65 | 43.97 (8–201) | 38.49 (8–193.2) | 51.21 (11–201) | < 0.001 *** | 685 |
| Triglycerides (mg/dl) | 35–135 | 129.04 (34–1265) | 118.18 (34–492) | 156.74 (54–1265) | < 0.001 *** | 213 |
| Creatine kinase (CK) (U/l) | 0–188 | 150.92 (20–2662) | 137.78 (25–2662) | 168.97 (20–1322) | 0.03 * | 216 |
| ALT/SGPT (%) | 0–33 | 25.79 (2–395.5) | 24.9 (5–395.5) | 27.32 (2–228.5) | 0.01 ** | 691 |
| AST/SGOT (U/l) | 0–31 | 30.33 (7–503) | 29.79 (7–503) | 31.26 (9–115) | < 0.001 *** | 462 |
| LDH (U/l) | 60–190 | 277 (110–2290) | 245.48 (117–727) | 319.81 (110–2290) | < 0.001 *** | 382 |
| Troponin T (TnT) (ng/l) | 0–14 | 20.32 (2–393) | 11.55 (2–162) | 32.79 (2–393) | < 0.001 *** | 213 |
| C-reactive protein (mg/l) | 0–5 | 50.16 (0–410) | 31.89 (0–206.1) | 74.26 (0.2–410) | < 0.001 *** | 603 |
| IL-6 (pg/ml) | 0–7 | 25.64 (1.72–166) | 7.17 (1.72–25.6) | 99.5 (33–166) | 0.04 * | 10 |
| Prothrombin Index (%) | 70–140 | 93.58 (4–137) | 96.45 (6–136) | 89.72 (4–137) | 0.02 * | 539 |
| Fibrinogen (mg/dl) | 200–400 | 530.15 (266–780) | 473.67 (266–780) | 600.75 (426–765) | 0.01 ** | 27 |
| D-dimer (ng/ml) | 0–500 | 1351.25 (180–21000) | 861.93 (180–12650) | 2010.07 (270–21000) | < 0.001 *** | 359 |
| Ferritin (ng/ml) | 15–150 | 426.71 (6.7–4557.3) | 317.18 (6.7–2100) | 629.86 (20.4–4557.3) | < 0.001 *** | 354 |
| Procalcitonin (ng/ml) | 0–0.5 | 0.36 (0.01–18.51) | 0.17 (0.01–16.8) | 0.59 (0.02–18.51) | < 0.001 *** | 349 |
| Lactate (mmol/l) | 0.3–2 | 1.37 (0.5–6) | 1.19 (0.5–3.9) | 1.5 (0.6–6) | < 0.001 *** | 268 |
| Platelets ($10^3$/μl) | 140–400 | 215.69 (12–729) | 221.79 (44–729) | 206.29 (12–560) | < 0.001 *** | 874 |
| Leukocytes ($10^3$/μl) | 3.8–10 | 7.07 (0.03–77.44) | 6.56 (1.41–54.32) | 7.86 (0.03–77.44) | < 0.001 *** | 875 |
| Neutrophils ($10^3$/μl) | 1.6–7.5 | 4.5 (0–18.81) | 4.01 (0.61–14.21) | 5.25 (0–18.81) | < 0.001 *** | 875 |
| Neutrophils (%) | 40–75 | 63.16 (0–95.6) | 60.28 (7.3–90.5) | 67.59 (0–95.6) | < 0.001 *** | 875 |
| Eosinophils ($10^3$/μl) | 0–0.6 | 0.12 (0–1.85) | 0.13 (0–1.85) | 0.1 (0–1.36) | < 0.001 *** | 875 |
| Eosinophils (%) | 0.5–7 | 1.75 (0–16.9) | 2.01 (0–16.9) | 1.35 (0–14.6) | < 0.001 *** | 875 |
| Basophils ($10^3$/μl) | 0–0.2 | 0.03 (0–0.16) | 0.03 (0–0.13) | 0.03 (0–0.16) | 0.02 * | 875 |
| Basophils (%) | 0–1.5 | 0.45 (0–2.1) | 0.48 (0–2.1) | 0.4 (0–1.8) | < 0.001 *** | 875 |
| Immature granulocytes ($10^3$/μl) | 0–0.03 | 0.04 (0–0.97) | 0.03 (0–0.29) | 0.06 (0–0.97) | < 0.001 *** | 875 |
| Immature granulocytes (%) | 0–0.5 | 0.55 (0–23.2) | 0.42 (0–8.5) | 0.75 (0–23.2) | < 0.001 *** | 875 |
| Lymphocytes ($10^3$/μl) | 0.9–3.5 | 1.82 (0.01–65.51) | 1.79 (0.26–48.29) | 1.86 (0.01–65.51) | < 0.001 *** | 875 |
| Lymphocytes (%) | 19–48 | 25.33 (0.9–88.9) | 27.77 (3.5–88.9) | 21.58 (0.9–86) | < 0.001 *** | 875 |
| Monocytes ($10^3$/μl) | 0.2–0.9 | 0.58 (0.02–7.36) | 0.57 (0.12–1.9) | 0.6 (0.02–7.36) | - | 875 |
| Monocytes (%) | 3.05–12 | 8.96 (0.5–66.7) | 9.18 (1.7–38.6) | 8.63 (0.5–66.7) | < 0.001 *** | 875 |
| Partial pressure of oxygen (mmHg) | 83–108 | 73.18 (14–271) | 81.73 (27–154) | 68.53 (14–271) | < 0.001 *** | 321 |
| Oxygen saturation (%) | 95–99 | 92.94 (21.2–100.8) | 96.66 (46.5–99.9) | 90.92 (21.2–100.8) | < 0.001 *** | 318 |
| Lymphocyte to Immature Granulocyte ratio | | 84.97 (0.75–889) | 101.26 (2.75–889) | 59.94 (0.75–430) | < 0.001 *** | 875 |

of around 60% for the three models, being the Top-5 the best model, with an accuracy of 67.33%. The Top-5 model also achieved the best confusion matrix, the best F1-score (0.68 for severe and 0.70 for Non-severe) and ROC curve (**Fig 4**).

## Discussion

Our study showed higher severity, ICU admission and death rates among male patients (**Table 1**), in accordance with previously reports [11]. The age distribution of severe cases followed a very similar pattern to that reported in the Italian population, with an increasing severity with age and a maximum peak of severe cases in octogenarians, followed by a decrease of this percentage in nonagenarians. Besides, the ICU occupancy (critical patients) was also similar,

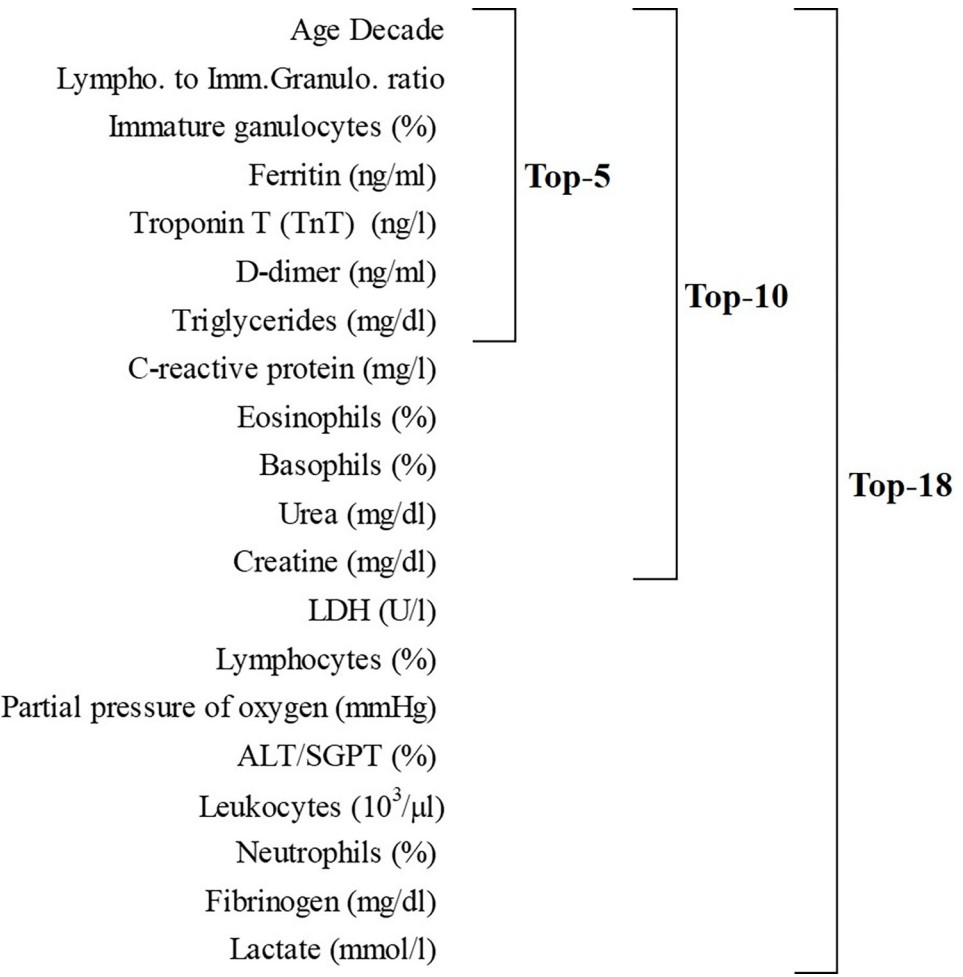

**Fig 3. List of the variables of the three classifications models.**

resulting in a 5.5% in our analysis and a 5.3% in the Italian study. The mortality rate was also analogous to previous reports, which was 12.94% in the present study, close to the 11% estimated for the Spanish population in the literature [2].

From the studied variables, only the absolute number of monocytes did not show significant differences between groups (**Table 3**). The analytical alterations observed in the biochemistry of COVID-19 patients are the reflection of the degree and the severity of multi-organ involvement [12], similar to what is observed in septic conditions of bacterial origin. These conditions commonly appear due to endothelial dysfunction and abnormal activation of the coagulation cascade, whose origins and mechanism are not completely understood yet [13].

Our results highlight the importance of use AI algorithms that offer the opportunity to identify new association that could be used as good predictors even when they are within normal ranges.

As reported in COVID-19, the cytokine storm is largely responsible for disease severity and it can have lethal outcomes [14]. Accordingly to previous reports, in our cohort, the percentage of lymphocytes showed statistically significant lower levels in severe compared to non-severe patients. Other parameters such as platelets, leukocytes, neutrophils, eosinophils, basophils and the percentage of monocytes showed statistically significant differences between severe

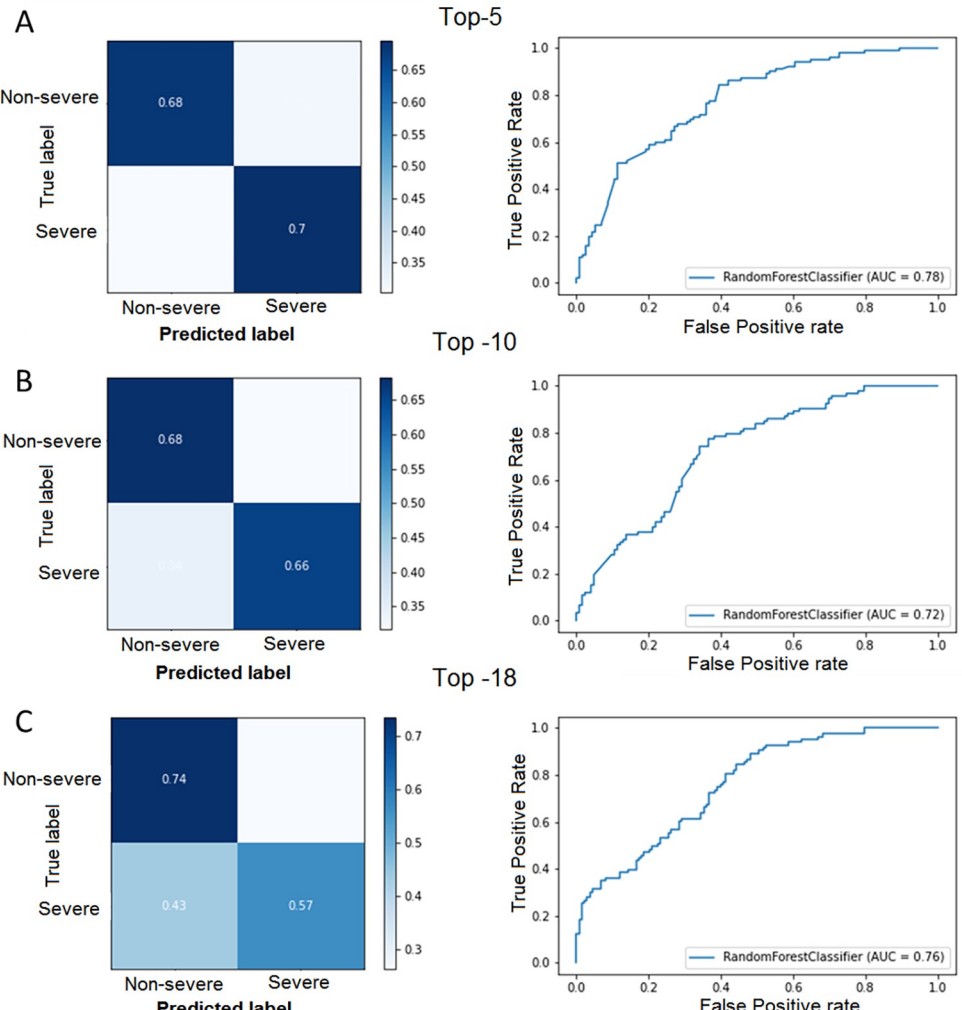

**Fig 4. Classification models for the prediction of disease severity.** The confusion matrix and ROC curves for the (**A**) Top-5, (**B**) Top-10 and (**C**) Top-18 models are shown.

and non-severe patients. These results are in accordance with what is reported in the literature [15]. Machine learning approaches have proposed as important variables in COVID19 severity prediction hyperferritinaemia, hypocalcaemia, hypoxia, hypoxemia and high levels of lactate dehydrogenase among others [16] and the use of metabolites as more powerful biochemical variables [17]

A parameter that was initially little explored in COVID-19 disease and that has shown to have prognostic implications is the number of immature granulocytes (IG) [18, 19], suggesting that the dysregulation of the myeloid compartment plays a role in the physiopathology of the COVID-19 disease. IG is the variable that show the greatest discriminatory weight, between those chosen by the AI algorithm. The percentage of immature granulocytes give us an approach to the presence of a higher production of myeloid-derived immature cells with suppression capacity (MDSCs). Interestingly, augmented MDSCs have been found in COVID-19 patients with a severe disease course [20–22], and it has been proposed that MDSCs expand their numbers under inflammatory stimuli [23] and that they can explain, at least in part, processes such as L-arginine mediated lymphopenia [24].

Therefore, the observed inverse relation between lymphocytes and immature myeloid cells, the positive relation between absolute numbers of IG and IL-6, and the physiopathological basis that explains, in part, the lymphopenia, led us to calculate the Lymphocyte to Immature Granulocyte ratio. Importantly, we found that this indirect measure of systemic inflammation was significantly decreased in patients that developed a severe COVID-19 disease when compared to the ones that only suffered a non-severe disease (Table 3). In line with our results, other authors found the immature neutrophil-to-VD2 (or CD8) T-cell ratio as an early marker for severe COVID-19 disease, predicting pneumonia and hypoxia onset. Moreover, it has been also reported, among other variables, a differential immature granulocyte to lymphocyte ratio between critical and non-critical COVID-19 patients, reinforcing the relevance of this measure [25].

The model with just 5 variables shows an AUC = 0.78 and a F1-score = 0.69. that is near to some of the previously presented AUC to predict severity [16, 17] showing promising utility as a clinical tool for predicting disease severity.

In conclusion, the present work highlights the importance of the study of analytic variables in the early phases of hospitalization by an easy and standard cost-effective technique present in all the hospitals. Our results also underline the benefits of applying AI methods to identify the best variables, even although they present values in normal ranges, to identify the best classifier. The prompt identification of the risk group is essential to thoroughly follow them and implement the necessary analytical and clinical interventions.

## Supporting information

**S1 File. Supporting information, including one figure and raw data.**
(ZIP)

## Author Contributions

**Conceptualization:** Andrés Roncancio-Clavijo, Miriam Gorostidi-Aicua, Jose Antonio Iribarren, Diego Clemente, Alvaro Prada, David Otaegui.

**Data curation:** Andrés Roncancio-Clavijo, Miriam Gorostidi-Aicua, Ainhoa Alberro, Andrea Iribarren-Lopez, Javier Basterrechea, Bruno Martinez, Alvaro Prada.

**Formal analysis:** Miriam Gorostidi-Aicua, Ainhoa Alberro, Andrea Iribarren-Lopez, Ray Butler, Raúl Lopez, Jose María Marimon, Alvaro Prada, David Otaegui.

**Funding acquisition:** David Otaegui.

**Investigation:** Andrés Roncancio-Clavijo, Miriam Gorostidi-Aicua, Ainhoa Alberro, Jose Antonio Iribarren, David Otaegui.

**Methodology:** Miriam Gorostidi-Aicua, Ainhoa Alberro, Jose María Marimon, Bruno Martinez, Alvaro Prada, David Otaegui.

**Project administration:** David Otaegui.

**Resources:** Javier Basterrechea, David Otaegui.

**Software:** Ray Butler, Raúl Lopez, Bruno Martinez.

**Supervision:** Ray Butler, Diego Clemente.

**Writing – original draft:** Miriam Gorostidi-Aicua, Jose Antonio Iribarren, David Otaegui.

**Writing – review & editing:** Andrés Roncancio-Clavijo, Miriam Gorostidi-Aicua, Andrea Iribarren-Lopez, Ray Butler, Raúl Lopez, Jose Antonio Iribarren, Diego Clemente, Jose María Marimon, Alvaro Prada, David Otaegui.

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
