## [Decision Letter · Decision Letter 0]

17 Aug 2022

PONE-D-22-16300Early biochemical analysis of COVID-19 patients helps severity predictionPLOS ONE

Dear Dr. Otaegui,

Thank you for submitting your manuscript to PLOS ONE. After careful consideration, we feel that it has merit but does not fully meet PLOS ONE’s publication criteria as it currently stands. Therefore, we invite you to submit a revised version of the manuscript that addresses the points raised during the review process. We have seen citation suggestions in the reviews. Remind you that this is not mandatory and that the inclusion or not of citations will have no influence on the decision made about your manuscript.

We look forward to receiving your revised manuscript.

Kind regards,

Kovy Arteaga-Livias

Academic Editor

PLOS ONE

Journal Requirements:

"The funders had no role in study design, data collection and analysis, decision to publish, or preparation of the manuscript"

5. Please ensure that you include a title page within your main document. You should list all authors and all affiliations as per our author instructions and clearly indicate the corresponding author.

Reviewers' comments:

Reviewer's Responses to Questions

**Comments to the Author**

1. Is the manuscript technically sound, and do the data support the conclusions?

Reviewer #1: Yes

Reviewer #2: Partly

2. Has the statistical analysis been performed appropriately and rigorously? 

Reviewer #1: Yes

Reviewer #2: No

3. Have the authors made all data underlying the findings in their manuscript fully available?

Reviewer #1: Yes

Reviewer #2: Yes

4. Is the manuscript presented in an intelligible fashion and written in standard English?

Reviewer #1: Yes

Reviewer #2: No

5. Review Comments to the Author

Reviewer #1: Overall, the study was very well conducted. I start by congratulating the authors. Below are my comments and I hope they are useful to improve your study.

1. Line 63, “Continuous variables have been described by the mean, minimum and maximum data”. Clinical data have almost always been shown not to have a normal distribution, my suggestion is that these results be presented in the form of median and interquartile range.

2. Line 73, “Statistical significance was defined as follows: p<0.001***”. Why are the levels of significance different? Was there any sampling procedure that was done to get to this point?

3.On the lines 99-100, "The Model Training step is 100 described in Figure 1". I saw to figure 1, it's actually just a graph of the age distribution as a function of severity. I understood nothing. Are you sure it's figure 1?

4.Lines 100-101, “Finally, a Cross-Validation 102 method is applied to decrease overfitting impact”. What type of cross-validation was used? Venetian blind?, leave-one-out?

5.I encourage the authors to insert the formulas for calculating the metrics for evaluating the performance of the models: sensitivity, specificity, accuracy(material and methods). Still on the evaluation of the performance of the models, I ask that the authors discuss the following scientific articles: PMID: 35751188, PMID: 34091385.

6.Still on the evaluation of the performance of the models, I ask that authors present a table of confusion matrix with: true positives, true negatives, false positives, false negatives, sensitivity, specificity and accuracy. This must be shown for all classifiers used. For more understanding, see the PMID paper table: 35751188.

7.What types of preprocessing are used for machine learning data? I encourage authors to clarify, in the material and methods section and in the Results section.

Reviewer #2: The article needs revision and re-writing with good language and grammar. 

1. What is the novelty of this research?

2.What did the authors find new ?

3. The data is old and dates back to the year 2020, and much research has been published on this topic.

4. Why did the authors not focus on albumin levels, despite the fact that albumin was found to decrease in much of the research

5.Why did the authors not focus on LDH levels, despite the fact that albumin was found to increase in much of the research.

6.Most  of the discussion is repetition of the result. Please delete and explore the result through analysis of the result 

In addition to some minor revisions.

Line 25, Please revise "COVID-19 is an infectious disease" as "Coronavirus Disease 2019 (COVID-19) is zoonotic infectious disease"

Please revise COVID-19 disease with "COVID-19" infection.

Line 8, delete "our," , because it globally effects the capacity protocol of hospitals.

On Line 9, add " AUC" to Intensive Care Units between brackets.

Lines 25-29, please add references. "Coronavirus disease 2019 (COVID-19) Situation Report – 57, WHO, 2020, 10 AM CET 17 March 2020"

6. PLOS authors have the option to publish the peer review history of their article (what does this mean?). If published, this will include your full peer review and any attached files.

Reviewer #1: **Yes: **Alexandre de Fátima Cobre

Reviewer #2: No

---

## [Author Response · Author response to Decision Letter 0]

31 Oct 2022

Dear Dr. Arteaga-Livias

Thanks for the opportunity to answer the reviewer’s comments. We are grateful for their carefully reading of our work and for their helpful advices and comments. We answer point-by-point each comment ad rewrite the manuscript incorporating these changes. We make an effort in explain better the algorithm and the used methodology an also the novelty and the focus of this work 

REVIEWER 1:

Reviewer #1: Overall, the study was very well conducted. I start by congratulating the authors. 

We thank the reviewer for their comment and their help to improve the manuscript

Below are my comments and I hope they are useful to improve your study.

1. Line 63, “Continuous variables have been described by the mean, minimum and maximum data”. Clinical data have almost always been shown not to have a normal distribution, my suggestion is that these results be presented in the form of median and interquartile range.

We totally agree with the reviewer´s suggestion, data presentation has been changed in the new manuscript version.

2. Line 73, “Statistical significance was defined as follows: p<0.001***”. Why are the levels of significance different? Was there any sampling procedure that was done to get to this point?

Our impression is that there is some misunderstanding. The type I level of significance has been set at 0.05, however in the charts, one star means that the p-value is below 0.05, two stars mean that is below 0.01, and three stars that is below 0.001. We will rewrite it in the new manuscript version.

3.On the lines 99-100, "The Model Training step is 100 described in Figure 1". I saw to figure 1, it's actually just a graph of the age distribution as a function of severity. I understood nothing. Are you sure it's figure 1?

Sorry about this mistake, as you notice there is an error in the order of the uploaded figures. Figure 1 is indeed a schematic description of the model.

4.Lines 100-101, “Finally, a Cross-Validation 102 method is applied to decrease overfitting impact”. What type of cross-validation was used? Venetian blind?, leave-one-out?

We use k-fold as a cross-validation method. This procedure has been used with a k =10 meaning a 10-fold cross validation. This information has been added to the new version of the manuscript.

5.I encourage the authors to insert the formulas for calculating the metrics for evaluating the performance of the models: sensitivity, specificity, accuracy (material and methods). Still on the evaluation of the performance of the models, I ask that the authors discuss the following scientific articles: PMID: 35751188, PMID: 34091385.

The formulas used are the ones by default in the used approach (SciPy):

- Sensitivity: True positives /(True positives+ False Negatives)

- Specificity: True Negative / (True negative + False Positives)

- accuracy: (True positive + True Negative) / (True Positive + True Negative + False Positive + False Negative)

We include in the discussion the proposed references.

6.Still on the evaluation of the performance of the models, I ask that authors present a table of confusion matrix with: true positives, true negatives, false positives, false negatives, sensitivity, specificity and accuracy. This must be shown for all classifiers used. For more understanding, see the PMID paper table: 35751188.

A new supplementary figure including the Table with the values for each model has been generated and added as Supplementary information.

7.What types of preprocessing are used for machine learning data? I encourage authors to clarify, in the material and methods section and in the Results section.

The dataset was constructed with the 42 features of 1082 patients, from these 37 variables were chosen avoiding redundancy. Aassociation and correlation between the variables were analyzed (AutoDiscovery Strength Score) to avoid model overfitting and to identify the patient's features that were more associated with the predictive goal. In this specific case, the association of the variables with the severity variable was evaluated and one of the correlated pair of variables was removed, if any. To do that, AutoDiscovery selected the proper numerical method based on the data type and distribution of the variables assessed. Numerical tests applied in each case are Spearman’s Rank Correlation, Variance Analysis (ANOVA one-way, U Mann-Whitney, and Kruskal-Wallis; normality was tested in these cases with D’Agostino/Pearson methods), and Cramer’s V Contingency Index. This massive data exploration process was carried out systematically for each patient group (stratum), considering only those correlations evaluated in a data subset with n≥5, two-tailed p < 0.05 and Spearman’s rank correlation coefficient > 0.8 to be relevant. 

Given the nature of this multiple testing method, a high significance threshold was calculated based on Benjamini-Hochberg method (False-Discovery Rate) to classify the rest associations. 

The Feature Selection process, where, based on Recursive Feature Elimination (RFE) and the AutoDiscovery Strength Score, age decade and the Lymphocytes to Immature Granulocyte ratio were added as variables for the model. 

This information has been added to the manuscript.

 

REVIEWER 2:

The article needs revision and re-writing with good language and grammar. 

Thanks to the reviewer for his/her comments that help us to improve the manuscript

1. What is the novelty of this research?

Although research in COVID19 moves very fast and the needs are changing, our work present6 an algorithm based in data that are obtained in all Hospitals routinely and no need for any extra analysis. That is novel and helpful for all the hospitals in the different countries to take fast decisions. Moreover our work underline the importance of the immature granulocites status in the gravity of the evolution.

2.What did the authors find new ?

We present an algorithm based in routinely parameters obtained in the clinical routine. Therefore decisions can be made in the first days of hospitalization without any additional analysis, just based in the regular data obtained the first day of hospitalization. We also highlight the importance of immature granulocytes in the evolution of the disease.

3. The data is old and dates back to the year 2020, and much research has been published on this topic.

We partially agree with the reviewer. We try to share our data as quickly as we could (and for that reason manuscript was located at BioRiv repository), however although the data are from the 2020, this is the most naïve data (without vaccinated people) and the ones that give us a more accurate idea about how the disease work. That could be useful for a new coronavirus pandemic and also to understand better the new cases that re arriving to the hospitals. 

4. Why did the authors not focus on albumin levels, despite the fact that albumin was found to decrease in much of the research.

The aim of the work was not to select the biomarkers but to use the ones that are routinely produced in the regular clinical practice. In fact we recover the data from the clinical records with no interaction (at that moment) with the clinicians, meaning that the choice of variables was based in the clinical needs and in the regular protocols. Is an undirected approach with the aim to find an easy and cheap way to make decisions at the hospital.

5.Why did the authors not focus on LDH levels, despite the fact that albumin was found to increase in much of the research.

Please, see response to answer 4.

6.Most of the discussion is repetition of the result. Please delete and explore the result through analysis of the result 

Thanks to the reviewer for his/her suggestions. We rewrite the discussion focusing in the analysis of the results.

---

## [Decision Letter · Decision Letter 1]

24 Nov 2022

PONE-D-22-16300R1Early biochemical analysis of COVID-19 patients helps severity predictionPLOS ONE

Dear Dr. Otaegui,

Thank you for submitting your manuscript to PLOS ONE. After careful consideration, we feel that it has merit but does not fully meet PLOS ONE’s publication criteria as it currently stands. Therefore, we invite you to submit a revised version of the manuscript that addresses the points raised during the review process.

We look forward to receiving your revised manuscript.

Kind regards,

Kovy Arteaga-Livias

Academic Editor

PLOS ONE

Reviewers' comments:

Reviewer's Responses to Questions

**Comments to the Author**

1. If the authors have adequately addressed your comments raised in a previous round of review and you feel that this manuscript is now acceptable for publication, you may indicate that here to bypass the “Comments to the Author” section, enter your conflict of interest statement in the “Confidential to Editor” section, and submit your "Accept" recommendation.

Reviewer #1: (No Response)

Reviewer #2: All comments have been addressed

2. Is the manuscript technically sound, and do the data support the conclusions?

Reviewer #1: Yes

Reviewer #2: Yes

3. Has the statistical analysis been performed appropriately and rigorously? 

Reviewer #1: No

Reviewer #2: Yes

4. Have the authors made all data underlying the findings in their manuscript fully available?

Reviewer #1: Yes

Reviewer #2: Yes

5. Is the manuscript presented in an intelligible fashion and written in standard English?

Reviewer #1: Yes

Reviewer #2: Yes

6. Review Comments to the Author

Reviewer #1: Dear Editor, I would like to thank you for the opportunity to review the article “Early biochemical analysis of COVID-19 patients helps severity prediction”, and I would like to take this opportunity to apologize for the delay in completing the review, I had unforeseen personal problems. Second, I would like to congratulate the authors for the article. Overall I enjoyed the post however some improvements need to be made. My observations follow.

1.In the “Abstract” section, the authors describe “…The test has been done by standardized cost-effective technique available in all the hospitals”. If the method is being used in clinical practice, it means that the model has been deployed. If so, I strongly recommend that you provide the access link for your Machine Learning App. I am just saying that the authors need to be realistic, because if the App has not been developed for the developed models, I recommend that they remove the phrase “…The test has been done by standardized cost-effective technique available in all the hospitals” in the manuscript.

2.In the material and methods section, “Continuous variables have been described by the mean, minimum and maximum data…”. I suggest initially testing the normality of the data. If normality is proven, the results must be presented as mean and standard deviation, otherwise the results must be presented as median and interquartile range.

3.The data used in this study are unbalanced: severe cases 449 (41.5%); ICU admission 60 (5.55%); deceased 140 (12.94%). It is important to point out that for imbalance problems, the AUC is not a better metric to evaluate the performance of the classifiers. The most reliable metric is the “F1 Score”. I strongly recommend that authors report these data in the article, including citing them in the abstract.

4.Still in material and methods, the authors describe “…Finally, a Cross-Validation 102 method is applied to decrease overfitting impact”. It needs to be clear which type of cross-validation was adopted.

5.The figures are very blurred. I ask you to improve the resolution of the figures. I also ask you to increase the size of the X and Y axes of the figures.

Reviewer #2: The author response to all reviewer comment , and the manuscript developed to the level of publication

7. PLOS authors have the option to publish the peer review history of their article (what does this mean?). If published, this will include your full peer review and any attached files.

Reviewer #1: **Yes: **Alexandre de Fátima Cobre

Reviewer #2: No

---

## [Author Response · Author response to Decision Letter 1]

12 Dec 2022

Dear Dr. Arteaga-Livias

Thanks for the opportunity to answer this second round of reviewer’s comments. We are grateful for their carefully reading of our work and for their helpful advices and comments. We answer point-by-point each comment ad rewrite the manuscript incorporating these changes. 

REVIEWER 1:

Dear Editor, I would like to thank you for the opportunity to review the article “Early biochemical analysis of COVID-19 patients helps severity prediction”, and I would like to take this opportunity to apologize for the delay in completing the review, I had unforeseen personal problems. Second, I would like to congratulate the authors for the article. Overall I enjoyed the post however some improvements need to be made. My observations follow.

1.In the “Abstract” section, the authors describe “…The test has been done by standardized cost-effective technique available in all the hospitals”. If the method is being used in clinical practice, it means that the model has been deployed. If so, I strongly recommend that you provide the access link for your Machine Learning App. I am just saying that the authors need to be realistic, because if the App has not been developed for the developed models, I recommend that they remove the phrase “…The test has been done by standardized cost-effective technique available in all the hospitals” in the manuscript.

Sorry for the misunderstanding. The model has been developed from biochemical data obtained in a routine assay. This sentence wants to emphasize that this kind of data could be easily obtained in the vast majority of hospitals and therefore the methodology proposed in our work can be easily transferred to clinical practice. We change this phrase in the abstract to clarify the point.

2.In the material and methods section, “Continuous variables have been described by the mean, minimum and maximum data…”. I suggest initially testing the normality of the data. If normality is proven, the results must be presented as mean and standard deviation, otherwise the results must be presented as median and interquartile range.

Thanks for the suggestion. As requested by a reviewer we change the presentation of these data to mean, minimum and maximum because, according to the reviewer, is the way in which clinical data should be presented. Moreover we perform a Normality test (Shapiro- Wilk) to confirm that our the variables should be presented in this way.

3.The data used in this study are unbalanced: severe cases 449 (41.5%); ICU admission 60 (5.55%); deceased 140 (12.94%). It is important to point out that for imbalance problems, the AUC is not a better metric to evaluate the performance of the classifiers. The most reliable metric is the “F1 Score”. I strongly recommend that authors report these data in the article, including citing them in the abstract.

Thanks for the comment. In this work we do not design the groups but use the data from the “real world”, recovering the data from a real situation from our hospital during a period of time. Following your suggestion we add the F1-score value to the manuscript.

4.Still in material and methods, the authors describe “…Finally, a Cross-Validation 102 method is applied to decrease overfitting impact”. It needs to be clear which type of cross-validation was adopted.

We use k-fold as a cross-validation method. This procedure has been used with a k =10 meaning a 10-fold cross validation. This information has been added to the new version of the manuscript.

5.The figures are very blurred. I ask you to improve the resolution of the figures. I also ask you to increase the size of the X and Y axes of the figures.

Quality of the figures and Axes have been improved

REVIEWER 2:

The author response to all reviewer comment , and the manuscript developed to the level of publication

---

## [Decision Letter · Decision Letter 2]

9 Mar 2023

Early biochemical analysis of COVID-19 patients helps severity prediction

PONE-D-22-16300R2

Dear Dr. Otaegui,

We’re pleased to inform you that your manuscript has been judged scientifically suitable for publication and will be formally accepted for publication once it meets all outstanding technical requirements.

Kind regards,

Kovy Arteaga-Livias

Academic Editor

PLOS ONE

Additional Editor Comments (optional):

Reviewers' comments:

Reviewer's Responses to Questions

**Comments to the Author**

1. If the authors have adequately addressed your comments raised in a previous round of review and you feel that this manuscript is now acceptable for publication, you may indicate that here to bypass the “Comments to the Author” section, enter your conflict of interest statement in the “Confidential to Editor” section, and submit your "Accept" recommendation.

Reviewer #2: All comments have been addressed

2. Is the manuscript technically sound, and do the data support the conclusions?

Reviewer #2: Yes

3. Has the statistical analysis been performed appropriately and rigorously? 

Reviewer #2: Yes

4. Have the authors made all data underlying the findings in their manuscript fully available?

Reviewer #2: Yes

5. Is the manuscript presented in an intelligible fashion and written in standard English?

Reviewer #2: Yes

6. Review Comments to the Author

Reviewer #2: All comments has been addressed .

The manuscript improved to the level of publication.......................

7. PLOS authors have the option to publish the peer review history of their article (what does this mean?). If published, this will include your full peer review and any attached files.

Reviewer #2: No

---

## [Editor Report · Acceptance letter]

10 May 2023

PONE-D-22-16300R2 

Early biochemical analysis of COVID-19 patients helps severity prediction 

Dear Dr. Otaegui:

I'm pleased to inform you that your manuscript has been deemed suitable for publication in PLOS ONE. Congratulations! Your manuscript is now with our production department. 

Kind regards, 

on behalf of

Dr. Kovy Arteaga-Livias 

Academic Editor

PLOS ONE